# Current Applications of Liquid Biopsy in Gastrointestinal Cancer Disease—From Early Cancer Detection to Individualized Cancer Treatment

**DOI:** 10.3390/cancers15071924

**Published:** 2023-03-23

**Authors:** Paul David, Anke Mittelstädt, Dina Kouhestani, Anna Anthuber, Christoph Kahlert, Kai Sohn, Georg F. Weber

**Affiliations:** 1Department of Surgery, University Hospital of Erlangen, Friedrich-Alexander University (FAU) Erlangen-Nürnberg, 91054 Erlangen, Germany; 2Department of Surgery, Carl Gustav Carus University Hospital, 01307 Dresden, Germany; 3Fraunhofer Institute for Interfacial Engineering and Biotechnology IGB, 70569 Stuttgart, Germany; 4Deutsches Zentrum für Immuntherapie, University Hospital of Erlangen, Friedrich-Alexander University Erlangen-Nürnberg, 91054 Erlangen, Germany

**Keywords:** liquid biopsy, circulating tumor cells (CTCs), circulating tumor DNA (ctDNA), tumor exosomes, tumor-educated blood platelets (TEPs), organoids, gastrointestinal cancer

## Abstract

**Simple Summary:**

Gastrointestinal (GI) cancers are a common cancer, affecting both men and women, normally diagnosed through tissue biopsies in combination with imaging techniques and standardized biomarkers leading to patient selection for local or systemic therapies. Liquid biopsies (LBs)—due to their non-invasive nature as well as low risk—are the current focus of cancer research and could be a promising tool for early cancer detection and treatment surveillance, thus leading to better patient outcomes. In this review, we provide an overview of different types of LBs enabling early detection and monitoring of GI cancers and their clinical application.

**Abstract:**

Worldwide, gastrointestinal (GI) cancers account for a significant amount of cancer-related mortality. Tests that allow an early diagnosis could lead to an improvement in patient survival. Liquid biopsies (LBs) due to their non-invasive nature as well as low risk are the current focus of cancer research and could be a promising tool for early cancer detection. LB involves the sampling of any biological fluid (e.g., blood, urine, saliva) to enrich and analyze the tumor’s biological material. LBs can detect tumor-associated components such as circulating tumor DNA (ctDNA), extracellular vesicles (EVs), and circulating tumor cells (CTCs). These components can reflect the status of the disease and can facilitate clinical decisions. LBs offer a unique and new way to assess cancers at all stages of treatment, from cancer screenings to prognosis to management of multidisciplinary therapies. In this review, we will provide insights into the current status of the various types of LBs enabling early detection and monitoring of GI cancers and their use in in vitro diagnostics.

## 1. Introduction

Gastrointestinal (GI) cancers are responsible for more cancer-related deaths than lung and breast cancer. Colorectal cancer (CRC) is the major type of GI cancer, with 1.9 million new cases diagnosed worldwide in 2020, making it after lung and breast cancer the third most common cancer of all organs. According to the International Agency for Research on Cancer, in the same year, 1.1 million new cases of gastric cancer, 900,000 new cases of liver cancer, 600,000 new cases of esophageal cancer, and 500,000 new cases of pancreatic cancer were diagnosed across the globe [1].

Although the prognosis of many GI cancers has improved over the past decades [2,3], a late cancer diagnosis is still the leading reason for cancer-related deaths among all GI cancers [4]. Current research focuses therefore on improving early cancer diagnosis, possibly leading to better outcomes among all GI cancers [5,6]. So far, endoscopic or CT-guided solid biopsies in combination with so-called serum-based tumor biomarkers are primary methods for the diagnosis of GI cancers [7]. Thereby, solid biopsies are considered the gold standard strategy capable of classifying tumors, identifying the mutational status, and providing prognostic information. However, these methods have some limitations, e.g., obtaining insufficient or inaccurate tissue samples possibly leading to false-positive or false-negative results. In addition, tissue biopsies might cause harm to the patient. However, recent studies suggest tissue biopsies taken from a single cancer nodule or single metastatic lesion may fail to represent the entire tumor heterogeneity within the patient, possibly being one of the main reasons for the failure of current targeted therapies [8,9,10,11,12,13]. To date, several serum-based biomarkers such as carcinoembryonic antigen (CEA), carbohydrate antigen 19-9 (CA19-9), carbohydrate antigen 72-4 (CA72-4), carbohydrate antigen 125 (CA125), and alpha-feto protein (AFP) have been identified and widely used for diagnosis, prognosis, and monitoring of potential recurrence of GI cancers [14,15]. Although, due to the limit of specificity and sensitivity most of these biomarkers are not useful for early cancer detection [16]. Therefore, LB emerged as a promising tool for early detection, treatment selection, and real-time prognosis.

In contrast to solid biopsy, LB is a minimally invasive approach enabling the real-time monitoring and early uncovering of alterations in cells or cell products shed from malignant lesions into the body fluids (Figure 1). LB analysis can identify multiple heterogeneous resistance mechanisms in single patients compared to solid biopsy. Furthermore, LB facilitates the choice of the right treatment and observation of the treatment response. Due to the minimally invasive nature of LB, the resulting complications from obtaining solid biopsies could be prevented. A typical LB sample is taken from any biological fluid such as blood, saliva, cerebrospinal fluid, or urine. LB materials derived from peripheral blood have been investigated extensively. LB analysis from blood contains enrichment and isolation of CTCs, circulating blood platelets, ctDNA, and other tumor genetic material such as extracellular vesicles. As of today, several LB technologies have been approved by the United States Food and Drug Administration (FDA) for malignancies such as metastatic lung, breast, prostate, or colorectal cancer: CELLSEARCH CTC test using circulating tumor cells from Veridex, Guardant360 CDx, and FoundationOne Liquid CDx using circulating cell-free DNA (cfDNA) and next-generation sequencing to detect tumor-specific mutations.

Due to the crucial role of LB markers, our main focus is on research findings and clinical applications in gastrointestinal cancers.

## 2. Overview of Different Methodologies and Their Current Clinical Application

### 2.1. Circulating Tumor Cells (CTCs)

CTCs are tumor cells, shed from a primary tumor. They can enter the bloodstream or lymphatic system, potentially spreading into distant organs possibly leading to metastases [17,18]. Nevertheless, only a minority of CTCs become solid metastatic lesions because of a complex sequence of events needed, i.e., the detachment from the primary tumor, migration through the circulating blood, immune escape, and survival. It remains unclear how the detachment process from the primary tumor tissue takes place. Evidence supports the involvement of epithelial to mesenchymal transition, by which transformed epithelial cells can acquire the ability to invade, resist apoptosis, and disseminate. This could be the main driver for the detachment of tumor cells from the primary tumor [19,20,21,22,23,24]. Other reports hypothesize that cells split into different clusters [25]. It is noteworthy that gastrointestinal cancers compared to breast cancer have lower numbers of CTCs in peripheral blood due to portal vein circulations and a steady ‘first-pass effect’ in the liver [26]. Therefore, portal vein blood might be a unique sample site to isolate CTCs from gastrointestinal cancers. It has already been shown that the number of enriched CTCs from portal vein blood is higher than in the systemic circulation [27,28]. Portal vein blood can be collected intraoperatively or even by endoscopic ultrasound (EUS)-guided sampling [29].

In addition to the number of CTCs, the analysis of physical (size, density, and electric charge) and biological (cell surface expression) properties could play a crucial role in future clinical use [30].

#### 2.1.1. Isolation and Enrichment of CTCs

The isolation and enrichment of CTCs are technically challenging because of their low numbers and elimination by the body’s immune system. The sensitivity of capture methods has improved in the last few years. First, based on cancer-specific characteristics, the cancer cells are separated from other blood components followed by enrichment procedures. There are two main techniques for isolating CTCs, one based on immunoaffinity properties, and the other based on biophysical properties. The immunoaffinity-based technology, including positive or negative selection assays, isolates CTCs with an antibody-immobilized inert surface combined with magnetic beads [31] (Figure 2a). The epithelial cell adhesion molecule (EpCAM) is a commonly used cell surface marker for positive CTC selection and an immunomagnetic assay called CELLSEARCH^®^. To date, CELLSEARCH^®^ is the only CTC technology that has gained FDA approval enabling the direct visualization and quantification of CTCs and the identification of living cells without the need for cell lysis. The functionality of CELLSEARCH^®^ for the detection of CTCs in GI cancer was confirmed by different studies [32,33]. In another approach, CTCs from blood samples of patients with CRC were pre-enriched through binding to VAR2CSA protein-coupled magnetic beads, and finally the colon-related mRNA transcripts USH1C and CKMT1A were detected by RT-qPCR [34]. Microfluidic chips as an immunoaffinity technology allow the selection of CTCs from small volumes of fluid under laminar flow, eliminating the need for sample processing. In a study by Lim et al. [35], single CTCs and CTC clusters were captured on the membrane of a centrifugal microfluidic device, picked without fixation, and used for further molecular analysis. Researchers began to develop CTC isolation technologies based on the biophysical properties of CTCs to overcome the bias and narrow spectrum of immunoaffinity-based approaches for CTC isolation. These methods are characterized as label-free and isolate CTCs from the blood based on biophysical properties, such as density, size, deformability, and electrical charge [36]. In order to confirm that the enriched cells consist only of CTCs, the cells must undergo characterization. Currently, this is achieved by immunocytochemistry-based assays, including immunofluorescence and immunohistochemistry, and molecular approaches, including RT-qPCR, FISH, and next-generation sequencing [37]. Using GILUPI Cell Collector (CC), a novel in vivo CTC detection device, researchers reported overcoming the limitations of small blood sample volumes. However, the clinical relevance of the CTCs detected was inferior to the CTCs identified by Cell Search [38].

#### 2.1.2. Clinical Application/Relevance of CTCs

Several studies have observed the prognostic value of CTCs for overall survival (OS) in localized colorectal cancer (CRC) [39,40]. In a series of 287 patients, a group demonstrated that preoperative CTC detection with a ≥ 1 CTC/7.5 mL proved to be an independent prognostic marker, whereas another group with 519 patients stated no association after surgery [39,41]. In high-risk CRC patients requiring adjuvant chemotherapy, CTC detection in the blood was correlated with worse outcomes [41,42,43]. A meta-analysis containing 1847 patients (11 studies) indicated that the detection of CTCs in the peripheral blood with CELLSEARCH^®^ has predictive utility for patients with CRC [44]. VISNÚ-1, a multicentre, randomized phase III trial with 349 patients, indicated that the first-line FOLFOXIRI-bevacizumab chemotherapy regimen significantly improved progression-free survival (PFS) in comparison to the FOLFOX-bevacizumab chemotherapy regimen in patients with metastatic CRC. This trial revealed that the CTC count might be a valuable non-invasive biomarker to aid in decision-making for patients undergoing intensive first-line therapy [45]. In a series of studies performed in metastatic CRC, patients with high baseline CTC count (≥3 CTCs/7.5 mL) could benefit from intensive chemotherapy regimens (four drugs), unlike patients with low CTC counts [44,46,47,48]. The PRODIGE 17 trial, conducted in 106 untreated patients with advanced gastric and esophageal cancer, reported that dynamic changes in CTC counts between baseline and 28 days after treatment were significantly associated with PFS and OS and could help in tailoring treatment regimens to each individual patient [49]. The above-mentioned findings might therefore help in the development of individualized treatment approaches in gastrointestinal cancer patients. In Table 1, a broad application of CTCs and its clinical relevance in GICs is mentioned.

### 2.2. Circulating Tumor DNA (ctDNA)

A group of French scientists detected cfDNA fragments in the circulating plasma of patients with autoimmune disorders in 1948 [73]. Later it became clear that the release of cfDNA is not restricted to autoimmune disorders, but was also found amongst others in pregnant women [74], septic patients [75], people suffering from different types of cancers, and even in healthy individuals [76]. Numerous studies have been conducted to uncover the mechanisms by which DNA fragments are released from cells into the plasma or serum. Major mechanisms involve apoptosis, necrosis, phagocytosis, NETosis, or active secretion [77]. Basically, every cell and tissue type is able to release cfDNA into circulation. Consequently, based on cell and tissue-specific methylation patterns from comprehensive databases including the Cancer Genome Atlas (TCGA) an assignment of cfDNA to different origins became possible [78]. Accordingly, the major source of cfDNA in blood results from hematopoietic cells, followed by vascular endothelial cells (up to 10%). However, cfDNA from liver tissue can also frequently be detected at low levels in circulation (up to 1%) in healthy people [78]. CtDNA is a fraction of cfDNA that originates from primary tumors, metastases, or from CTCs. Additionally, some findings are suggestive of an active release of ctDNA from living tumor cells involving exosomes [79]. CtDNA is characterized by small 70–200 base pair fragments circulating freely within the blood [73] showing a major fragment size of around 170 base pairs, which corresponds to nucleosomal fragments resulting from apoptosis [80]. The half-life of ctDNA is very short ranging from 15 min to 2.5 h before it is finally cleared by the liver and/or kidneys which is a prerequisite for a precise biomarker [81]. Concentrations of ctDNA in the blood of patients with a malignant tumor are significantly increased compared to healthy individuals [82]; however, levels of released ctDNA significantly vary between different tumor types and tumor stages. Furthermore, the match of cancer-specific alterations in the genome of solid tumors to those of ctDNA is a major discriminator between ctDNA and physiological cell-free DNA at steady state [83,84,85].

#### 2.2.1. Detection and Analysis of ctDNA

There are a number of technical challenges associated with the analysis of ctDNA. First, total cfDNA itself is present only at comparably low concentrations in the nanogram per ml range. Second, the fraction of ctDNA among total cfDNA in many cases is also relatively low. This becomes especially evident in the early stages of cancer development or for tumors that release only low amounts of DNA [76]. Therefore, an urgent need for enrichment approaches of ctDNA over total cfDNA still exists. In this context, it has been reported that ctDNA to some extent might be enriched in cfDNA fractions of smaller size (90–150 base pairs (bp)) separated from the major fraction of 170 bp fragments [86] (Figure 2b). Although a tendency to higher ctDNA content in the 90–150 bp fraction could be found, enrichment factors of twofold in more than 95% of cases were still not satisfying [86]. In principle, a plethora of downstream analyses has been established for the diagnosis, prognosis, monitoring, and prediction of treatment response for all major cancer diseases. Among them are single nucleotide polymorphism (SNP) as well as copy number variation (CNV) analyses. When sufficient amounts of ctDNA templates (at least one genome equivalent per assay) are available in patient specimens, targeted assays based on PCR, for example, represent powerful approaches. Such PCR-based techniques which are characterized by high sensitivity showed the ability to identify relevant alterations in genes including RAS, HER2/NEU, BRAF, MET, BRCA2, APC, TP53, ALK, ROS1, PTEN, and NF1. However, when ctDNA content drops below one tumor genome equivalent per assay multi-target approaches become mandatory. Multi-gene panels are well established to similarly test for hundreds of targets. However, when it comes to early detection of cancer for screening purposes ctDNA contents of less than 0.01% represent a serious technical challenge to robustly assure high sensitivity and specificity. Cancer-specific methylomes in ctDNA were therefore proposed as a promising alternative [87]. In combination with next-generation sequencing (NGS), thousands of differential hyper-/hypo-methylated regions (DMRs) have been determined for a variety of cancers in addition to chromosomal and copy number changes or point mutations [88,89]. Accordingly, complex signatures of DMRs, for example, allow for early diagnosis with higher reliability. In general, it has been suggested that even for only as low as 0.01% ctDNA content prediction of cancer disease should be possible with 100–1000 DMRs when sequencing coverage of at least 100–1000x is given [87].

#### 2.2.2. Clinical Application/Relevance of ctDNA

Analysis of tumor-linked genetic alterations and DNA methylation profiling has been recognized as a method for detecting potential biomarkers for disease diagnosis and prognosis [90]. In a study by Tam et al. [91] the levels of TAC1 and SEPT9 methylation detected in postoperative sera of patients with CRC were independent predictors for tumor recurrence and unfavorable cancer-specific survival. Findings from several studies showed that high ctDNA levels combined with an increased number of mutations detected in the ctDNA were linked to poor survival and multi-site metastasis [92]. However, when the amount of ctDNA was <1 mutant template molecule per milliliter of plasma, tests fail to detect early-stage cancer [93]. Therefore, methylome analyses of cfDNA with thousands of cancer-specific DMRs might overcome such limitations. In 2019, the ‘Galleri test’ or the ‘Galleri multicancer early detection (MCED) test’ developed by GRAIL Inc. (Menlo Park, CA, USA) achieved Breakthrough Device designation. The Galleri test detects cancer- and tissue-specific alterations in the methylation patterns of cfDNA in a blood sample via NGS, which should allow early pan-cancer detection even for cancers with unknown primary. GRAIL’s clinical trial includes three studies: the Circulating Cell-free Genome Atlas (CCGA) Study (Clinical Trial NCT02889978) [94], the STRIVE Study (Clinical Trial NCT03085888), and the SUMMIT Study (Clinical Trial NCT03934866). In these studies including 2482 cancer patients covering approximately 50 different cancer types, sensitivities and specificities were tested by using a target hybridization capture approach for the analyses of 100,000 differentially methylated regions [95]. In a predefined subset of 12 cancer types which comprised roughly 63% of all US cancer cases average sensitivity was 67.3% accumulated for stage I–III at a very high specificity of 99.3% [95]. Sensitivity for all 50 cancer types accumulated for stage I–III dropped to 43.9% with 18% sensitivity for stage I and 43% in stage II. Remarkably, sensitivities for pancreatic cancer were significantly higher even at early stages with 63% and 83% in stage I and II, respectively. Although high specificities might predestine this test for negative prediction in screening approaches, moderate sensitivities still require improvements, especially for early diagnosis. Another FDA-approved LB-based test is Epi proColon^®^ (Epigenomics AG, Berlin, Germany) targeting methylation changes used to screen for CRC. The test is based on a real-time PCR with a fluorescent hydrolysis probe and targets the methylation changes of the SEPT9 gene promoter in cfDNA isolated from plasma. To evaluate the clinical assessment, Epi proColon^®^ was involved in a prospective multicenter study (Clinical Trial NCT00855348) [96,97,98].

A ctDNA-guided approach has been used by numerous clinical trials (DYNAMIC, CIRCULATE-Japan, CIRCULATE-trial, CIRCULATE-PRODIGE, and IMPROVE-IT2), all focused towards precise adjuvant therapy for stage II colon cancer patients [99,100,101,102,103]. According to DYNMAIC-trial, a ctDNA-guided approach led to a reduction in the number of patients who received adjuvant therapy, and furthermore, ctDNA-positive patients appeared to benefit from adjuvant treatment. The other trials (CIRCULATE-trial, IMPROVE-IT2) mentioned above could help in decision-making before adjuvant treatment in stage II colon cancer. The outcome of these trials proposes that a survival benefit from adjuvant chemotherapy may be obtained in a well-defined subgroup of patients with stage II colon cancer—especially those with detectable ctDNA post-surgery. In the case of pancreatic cancer, ctDNA might be used as a marker for monitoring treatment efficacy and disease progression [104]. To prove the feasibility of a non-invasive detection in plasma, Liu et al. developed a pancreatic cancer detection assay (PANDA) for screening and validation of PDAC-specific DNA methylation in tissues and plasmas of PDAC patients [105]. In combination with age and CA19-9 plasma serum level, this assay showed encouraging results to discriminate PDAC plasma from non-malignant disease, showing its capability to be amended into a non-invasive diagnostics method for PDAC screening. In Table 2, a broad application of ctDNA/cfDNA and its clinical relevance in GICs is mentioned.

### 2.3. Circulating Extracellular Vesicles (Tumor Exosomes)

Exosomes are a subpopulation of extracellular vesicles (EVs), ranging in size from 30–150 nm. They are derived from the endosomal pathway via the formation of late endosomes or multivesicular bodies (MVBs). As an important mediator of intracellular communication, exosomes transmit various biological molecules including proteins, lipids, and nucleic acids over distances within the protection of a lipidic bilayer-enclosed structure. Nearly all types of cells and all body fluids contain exosomes [124,125]. Cancer cells and other stromal cells in the tumor microenvironment (TME) also release exosomes and control tumor development through molecular exchanges mediated by exosomes [126,127]. Circulating extracellular vesicles (cEVs) are implied to be more stable in comparison to serological proteins as the lipidic bilayers defend the content from proteases and other enzymes [128].

#### 2.3.1. Isolation of Tumor Exosomes

Exosomes can be isolated using various methodologies. Ultracentrifugation (UC) is the most widely used technique. Other techniques include differential centrifugation (DC), density gradient ultrafiltration (DG), size exclusion chromatography (SEC), precipitation, immunoaffinity capture based on the expression of endosomal surface proteins such as CD81, CD63, and CD9, and microfluidic-based assays [129] (Figure 2d).

#### 2.3.2. Clinical Application/Relevance of Tumor Exosomes

Exosomes demonstrate significant advantages over other sources of LBs. First, exosomes exist in almost all body fluids and are characterized by highly stable lipidic bilayers. Second, living cells secret exosomes. They thus contain biological information from the parental cells and are more representative than cell-free DNA secreted during necrosis or apoptosis [130]. Third, exosomes express specific proteins such as CD63, ALIX, TS101, and HSP70, 20 which can be used as markers to discriminate exosomes from other vesicles making their identification clear and simple [131]. Fourth, as exosomes can present specific surface proteins from parental cells or target cells, they can help in the prediction of organ-specific metastasis [132]. Fifth, compared to CTCs, they can be isolated using classic methods such as ultracentrifugation [133].

Circulating exosomal PD-L1 was shown to contribute to immunosuppression, to reflect the immune status, and to better predict survival in patients with GICs, thereby making it a potential prognostic biomarker [134]. Additionally, EV proteins such as carcinoembryonic antigen-related cell adhesion molecules (CEACAMs), Tenascin C, Glypican-1, and ZIP-4 have been recognized as diagnostic biomarkers in GICs [135,136,137]. The EV-Glypican-1 (GPC-1) derived from plasma has been described as a potential marker of early pancreatic ductal adenocarcinoma (PDAC) with higher diagnostic accuracy than CA19-9 [137]. A clinical trial (NCT03032913) performed by Etienne BUSCAIL involved 20 PDAC patients and 20 non-cancer patients, whose blood samples were collected to detect CTCs and GPC1+ exosomes for diagnostic accuracy assessment of CTCs and Onco-exosome Quantification. Very recently, Lin et al. presented the development of a signature of four EV-proteins—monocyte marker CD14, Serpin A4 (a regulator of angiogenesis), CFP (a positive regulator of the complement system), and LBP (lipopolysaccharide binding protein)—as prognostic biomarkers in colorectal liver metastases (CRLM). Thereby, they used matching pre- and post-operative serum samples of patients undergoing CRLM surgery and finally validated the discovered proteins in three independent cohorts. Additionally, they showed that EV-bound CXCL7 could serve as a biomarker of early response in CRML patients undergoing systemic chemotherapy [138]. Furthermore, three cancer-specific phospholipids were found in a study that analyzed 20 dysregulated phospholipids in pancreatic cancer compared to controls. Among them, LysoPC 22:0 was linked with tumor stage, whereas CA19-9 and CA242 were associated with tumor diameter and positive lymph node count [139]. Finally, diagnostic accuracy for WASF2, ARF6 mRNAs, SNORA74A, and SNORA25snoRNAs in circulating exosomes was greater than for CA19-9 in discriminating PC patients from controls [140]. Even though a limited number of studies on the application of exosome-based drug delivery vectors in the treatment of GICs exist, some of them report intriguing advancements in the field. Using an in vitro model Pascucci L. et al. demonstrated the role of exosomal-mesenchymal stromal cells (exo-MSCs) in the packaging and delivery of active drugs, suggesting a possible option of using the MSCs as a warehouse to develop drugs with a better specificity [141]. Another study showed the application of milk-derived exosomes for the oral delivery of PAC in early-stage and advanced-stage pancreatic and other cancers. It also evaluated the anti-tumor potency of the milk-derived exosomes loaded with PAC [142]. Furthermore, studies have verified the link between cancer-derived exosomes and the modulation of immune response in pancreatic cancer and have also indicated the application of these cargo carriers in targeting pancreatic cancer cells, whether as anti-tumor drugs and other molecules, such as RNAi against mutant KRAS [143]. More applications and the clinical relevance of exosomes are summarized in Table 3.

### 2.4. Tumor-Educated Blood Platelets (TEPs)

Platelets play a central role in blood coagulation and in the healing of wounds, and their relationship with cancer has been extensively investigated [156,157,158,159,160,161,162,163,164]. There are two major studies that indicated the involvement of platelets during tumor progression. These studies contributed to the development of the concept of tumor-educated platelets (TEPs). The first study by Trousseau (1868) observed spontaneous coagulation being common in cancerous patients, stating that circulating platelets were affected by cancer [165]. The second study (Billroth T, 1877) described ‘’ thrombi filled with specific tumor elements’’ as part of metastasis, pointing out a direct interaction of cancer cells and platelets [166,167]. In recent years, several studies have focused on the impact of platelets during cancer progression, and some of them showed that platelet dysfunction and thrombotic disorders are important key factors in cancer development. By now it is well known that tumor-educated platelets (TEPs) are educated when they interact with the tumor cells in such a way as to lead to the detachment of biomolecules, such as proteins and RNA, tumor-specific splice events, and finally to megakaryocyte alteration [168]. Through this interaction, the RNA profile of blood platelets changes. This change has been used as an independent diagnostic marker for detecting TEPs in various solid tumors [169]. The RNA biomarkers of the directly transferred transcripts are EGFRvIII, PCA3, EML4-ALK, KRAS, EGFR, PIK3CA mutants, FOLH1, KLK2, KLK3, and NPY [170]. Here, we describe the isolation, detection, and clinical relevance of TEPsin GICs.

#### 2.4.1. Isolation and Detection of Tumor-Educated Platelets

Tumor-educated platelets can be separated from peripheral blood by using a two-step centrifugation approach [171]. The first step separates the platelet-rich plasma (PRP) from the red blood and white blood cells, whereas the second centrifugation step yields the platelet pellet [172]. For the detection of the TEPs in the human plasma, the plasma pellets are dissolved in 1ml TRIZOL reagent. RNA is extracted from the plasma platelet pellet and the suboptimal quality of the platelet RNA is characterized by an absence of ribosomal 18S and 28S peaks and measured in a Bioanalyzer Picochip. Preferably, platelet mRNAs are measured by other methods, such as Fragment analyzer (Advanced Analytical Technologies, Ankeny, IA, USA) or Qubit Fluorometric Quantification (Thermo Fisher, Waltham, MA, USA) analysis (Figure 2c).

#### 2.4.2. Clinical Application/Relevance of Tumor-Educated Blood Platelets

Platelet-related measures are considered important in anticipating long-term results in patients with GI cancer. A study distinguished 228 patients with localized and metastasized tumors from 55 healthy donors using the genetic profile of mRNA from TEPs. Additionally, mRNA sequencing of TEPs could accurately recognize MET or ERBB2-positive and mutant KRAS, EGFGR, or PIK3CA tumors. Moreover, TEPs have the ability to identify the location of primary tumors including colorectal cancer, non-small lung carcinoma, glioblastoma, pancreatic cancer, hepatobiliary cancer, and breast cancer [169]. Yang et al. showed that TIMP metallopeptidase inhibitor 1 (TIMP1) mRNA levels were higher in platelets from patients with CRC compared to those from healthy donors or patients with inflammatory bowel diseases, which could be another promising diagnostic signature [173]. A study in patients with hepatocellular carcinoma (HCC) conducted by Asghar et al. showed that the expression of TGF-β, NF-κβ, and VEGF was increased in TEPs of HCC patients compared to that from controls and thereby they suggested that these RNA based biomarkers could be used as a promising tool for early detection of HCC [174]. Furthermore, the alterations of platelet counts as a prognostic marker were studied in a clinical trial (NCT03717519) conducted by Corrado Pedrazzani. The study recruited 196 patients with synchronous colorectal liver metastases. In esophageal squamous cell carcinoma (ESCC), Ishibashi et al. performed a meta-analysis evaluating the prognostic values of platelet-related measures. The analysis revealed that a high platelet-to-lymphocyte ratio (PLR) was significantly correlated with poor OS [175]. A dual-center retrospective study performed by Yang et al. showed a standardized indicator of platelet counts was used to forecast the prognosis of 586 CRC patients by using a development-validation cohort. In the development cohort, postoperative platelet count and postoperative/preoperative platelet ratio (PPR) were independent predictors of prognosis in CRC patients. In the validation cohort, the platelet/lymphocyte ratio and PPR were used to test the OS of CRC patients and showed the largest AUC in reviewing 1-year and 3-year OS (AUC: 0.663 and 0.673) [176]. Based on the studies above, we summarize that PLR and PPR could serve as reliable and economic indicators to evaluate the prognosis of GI cancer. Interestingly, many new approaches have been utilized to explore the clinical relevance of TEPs in GICs. We have summarized some of them in Table 4 below. TEPs have advantages over other blood-based sources due to their abundance in the blood, the ease with which they can be isolated, their high-quality RNA, and their ability to process RNA in response to foreign signals.

## 3. Conclusions

Due to the limitations of conventional tissue biopsies, there is an urgent need for new tumor biomarkers. As reviewed here, LBs have been well-evaluated and show promising results as an alternative clinical tool for the detection and treatment of gastrointestinal cancers, many of those currently being intensively investigated in various observational and interventional clinical trials (Table 5 and Table 6). Multiple longitudinal biopsies allow for real-time monitoring of the tumor. This approach may facilitate the prediction of the possible treatment outcome and may help in choosing the optimal individualized therapeutic strategy. Therefore, the analysis of LB markers (TEPs, CTCs, ctDNA/cfDNA, and exosomes), in combination with modern imaging techniques and already existing protein markers might help to create an optimal clinical synergy that might be used as a standard procedure in the near future for early cancer detection and individualized cancer treatment.

## Figures and Tables

**Figure 1 cancers-15-01924-f001:**
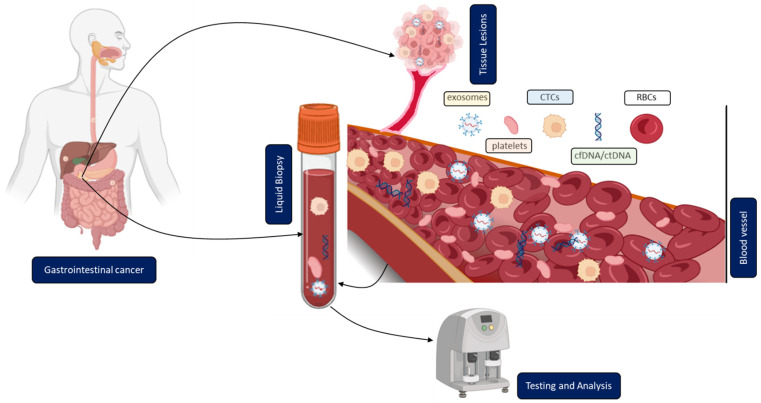
Clinical application of liquid biopsy (LB) in gastrointestinal cancer (GICs). Circulating tumor cells (CTCs), cell-free or circulating tumor DNA (cfDNA/ctDNA), tumor-educated platelets (TEP), exosomes, and RBCs in the blood of GICs patients can be used as potential biomarkers for LBs and their expression levels can be measured to determine the clinical status of GICs patients.

**Figure 2 cancers-15-01924-f002:**
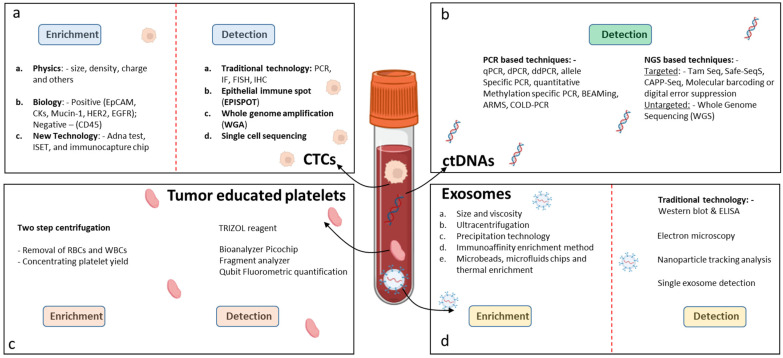
Techniques for detection of LB biomarkers in GICs. (**a**) Detection of CTCs, (**b**) detection of ctDNAs, (**c**) detection of tumor-educated platelets, and (**d**) detection of exosomes.

**Table 1 cancers-15-01924-t001:** Clinical relevance of CTCs in GICs.

Cancer Type	Threshold	Sample Size (Number)	Sensitivity	Specificity	AUC	Clinical Significance	References
Gastric cancer	2 CTCs	116	85.3	90.3	0.928	Distinguish between GC patients and healthy controls and provide clinical output	[50]
Gastric cancer	CTC-PD-L1	32 with progressive GCs				Monitor prognosis and guide future individualized immunotherapy	[51]
Gastric cancer	CSV+PD-L1+CTCs	70	71			Predicts treatment response and prognosis in GC patients	[52]
Gastric cancer	CTCs and TWIST	32 with metastatic cancer	80.6			As a prognostic marker	[53]
Gastric cancer	CTCs/cfDNA	45 patients with progressive GC	95.6			Predicting the efficacy and prognosis of neoadjuvant chemotherapy for progressive GC	[54]
Colorectal cancer	≥3 (chemotherapy and serum CEA)	121				Presence of CTCs might be valuable for predicting survival outcome	[55]
Non-metastatic colorectal cancer (NMCRC)	≥4 (CS. CK19, MUC1, CD44, CD133 and ALDH1)	63	68.3	95		CTCs could be novel therapeutic targets for NMCRC	[56]
Metastatic colorectal cancer (mCRC)	≥1.92 (CEACAM)	436				Detection of peripheral blood CEACAM5 mRNA-positive CTCs as an adverse prognostic factor correlated with poor clinical outcome in patients with mCRC	[57]
Duke’s stage B and C colorectal cancer	(carcinoembryonic antigen CEA), cytokeratin (CK) 19, CK20, and/or CD133 (CEA/CK/CD133)	735				CTC as a detection marker in patients with Duke’s stage B and C	[58]
Advanced CRC	≥3 CTCs (EpCAM, CK and CD45)	467				CTC count before and during chemotherapy treatment as an independent predictor of PFS and OS in advanced CRC patients	[59]
Colorectal cancer	≥3 CTCs (EpCAM, CK and CD45)	430				CTC count before and during chemotherapy treatment as an independent predictor of PFS and OS during metastatic CRC patients	[60]
Hepatocellular carconoma	≥5 CTCs EpCAM and mucin1	73				CTCs could possibly be a novel prognostic biomarker in HCC	[61]
Hepatocellular carconoma	≥2 CTCs, EpCAM, CD8/18/19	964				The CellSearch system could determine the clinical utility of CTCs in HCC	[25]
Hepatocellular carconoma	≥5 CTCs, Glypican-3	85				GPC-3 as a useful biomarker for HCC patient outcomes	[62]
Hepatocellular carconoma	EpCAM	299				CTC detection by qPCR could be utilized in clinics for auxiliary diagnosis, treatment response assessment, and decision making	[63]
Hepatocellular carconoma	ASGPR, Hep Par 1	85				A highly sensitive and specific CTC detection tool	[64]
Hepatocellular carconoma	(EpCAM)/vimentin/Glypican-3 (GPC3)	44	96.94	98.12		A convenient and feasible CTC capture system to predict clinical outcomes in HCC patients	[65]
Hepatocellular carconoma	PD-L1	87	71.1	91.8		Favorable response to anti-PD-1 therapy is associated with the presence of PD-L1+ CTCs	[66]
Hepatocellular carconoma	pERK+/pAkt− CTCs	109				pERK+/pAkt− CTCs are sensible to sorafenib	[67]
Pancreatic cancer	ISET	165				Higher CTC counts correlate with earlier recurrence. Increase in CTC numbers after neoadjuvant treatment. CTC+ correlates with early recurrence and OS in the pretreated group.	[68]
Pancreatic cancer	CK20	172				CTC predicts poor OS	[69]
Pancreatic cancer	CK20	25	88	90		CTCs predict the prognosis of pancreatic cancer	[70]
Pancreatic cancer	NanoVelcro CTC assay (CK)	100				CTC as a promising prognostic biomarker for PDAC patients	[71]
Pancreatic cancer	LIN28B	35				Molecular characterization of CTCs provides a unique opportunity to correlate gene set metastatic profiles, identify drivers of dissemination, and develop therapies targeting the “seeds” of metastasis	[72]

**Table 2 cancers-15-01924-t002:** Clinical relevance of ctDNA/cfDNA in GICs.

Cancer Type	Sample Size	Sensitivity (%)	Specificity (%)	AUC	Clinical Significance	References
Gastric cancer	46 patients with stage I–III GC	39	100		MRD with ctDNA testing identifies patients at high risk of recurrence	[106]
Gastric cancer	61 cases of partially metastatic GC				Associated with improved prognosis	[107]
Gastric cancer	1145	87		0.984	Potential to expand access to targetted therapies and immunotherapy to all patients with advanced cancer	[108]
Gastric cancer	428	68.9	95.8	0.98	Predicts response to chemotherapy and surgery in patients with CRC; tumor recurrence should be considered in GC with persistently elevated cfDNAs levels after surgery	[109]
Gastric cancer	124	78.96	91.81	0.94	For early screening of GC	[110]
Gastric cancer	30	96.67	94.11	0.991	For early detection of cancer and assessment of tumor load	[111]
Pancreatic cancer	39	97.3	91.6		Minimal invasive blood-based biomarker panel which could potentially be used as a diagnostic and screening tool in a select subset of high-risk populations	[112]
Pancreatic cancer	194				ctDNA in combination with exosomal DNA provides both predictive and prognostic information relevant to therapeutic stratification	[113]
Colorectal cancer	455				Reduced the usage of adjuvant chemotherapy	[99]
Colorectal cancer	250				Detection of residual disease	[114]
Locally advanced rectal cancer (LARC)	462				ctDNA analysis as a useful guide for adjuvant chemotherapy selection in LARC patients	[115]
Pancreatic cancer with liver metastasis	104				Use of circulating tumor DNA as an independent prognostic marker for advanced pancreatic cancer	[116]
Pancreatic cancer	135				ctDNA as an independent prognostic marker in advanced PDAC as well as an indicator of shorter disease-free survival in resected patients when detected after surgery	[117]
Pancreatic cancer	112				Increased ctDNA levels were a poor prognostic factor for survival.	[118]
Pancreatic cancer	259				Plasma cfDNA might provide a prognostic and diagnostic tool to assist surgical decision-making in PDAC patients	[119]
Pancreatic cancer	189				Longitudinal ctDNA KRAS assists in therapeutical decision-making and provides a kinetically robust and quantitative measurement of patient response.	[120]
Pancreatic cancer	171	86	88		ctDNA methylation approach to discriminate PDAC plasma from non-malignant diseases	[105,121]
Pancreatic cancer	101				ctDNA as genetic predictors of result in pancreatic cancer and might open new avenues of therapeutic intervention.	[122]
Pancreatic cancer	112				ctDNA-guided approach intensified the treatment strategies for pancreatic cancer patients.	[123]

**Table 3 cancers-15-01924-t003:** Clinical relevance of Exosomes in GICs.

Cancer Type	Biomarkers	Sample Type	Expression	Clinical Significance	References
Gastric cancer	miRNA-4741, miR-32, miR-3149 and miR-6727	tissue and plasma	miR-4741—upregulated miR-32, miR-3149 and miR-6727—downregulated	Acts as a diagnostic marker for GC and an influential factor in inhibiting GC progression	[144]
Gastric cancer	LncRNAH19GC	serum	downregulated	Possible biomarkers with diagnostic and prognostic value	[145]
Gastric cancer	hsa_circ_00115286	tissue, plasma, and cells	upregulated	Possibly a non-invasive biomarker for GC diagnosis and prognostic assessment	[146]
Gastric cancer	TRIM3	serum	downregulated	Inhibition of GC progression in vitro and in vivo	[147]
Gastric cancer	MET	cells	upregulated	Amplifies tumor growth and development in vitro and in vivo	[148]
Colorectal cancer	Exo-EpCAM	plasma	upregulated	May have potential as non-invasive biomarkers for detection of CRC	[149]
HCC/Colongiocarcinoma	EpCAM	serum	upregulated	A novel non-invasive biomarker to assess the presence and possible extent of cancers in patients with advanced liver disease	[150]
Esophageal cancer	Stathmin	serum	upregulated	A very promising diagnostic and predictive marker for SCC in the clinic, especially for ESCC	[151]
Colorectal cancer	CD147	blood	upregulated	EV-mediated intercellular communication and the development of advanced diagnostic and therapeutic strategies	[152]
Colorectal liver metastasis (CRLM)	CXCL17	serum	downregulated	EV-bound CXCL7 was found as a biomarker of early response in CRLM patients receiving systemic chemotherapy	[138]
HCC/Colongiocarcinoma	CD147	serum	upregulated	A novel non-invasive biomarker to assess the presence and possible extent of cancers in patients with advanced liver disease	[150]
Colorectal cancer	Hsp60	cells	upregulated	Biomarker for diagnostics, assessing prognosis, and monitoring disease progression and response to treatment, particularly in cancer	[153]
Colorectal cancer	Glypican-1 (GPC1)	plasma	upregulated	Specific markers for the diagnosis of CRC and targets for the therapy of CRC.	[154]
Colorectal cancer	CopineIII (CPNE3)	plasma	upregulated	Exosomal CPNE3 show potential implications in CRC diagnosis and prognosis.	[155]
Pancreatic cancer	CEACAMs	pancreatic fluid	upregulated	Exosome isolation is feasible from pancreatic duct fluid, and that exosomal proteins may be utilized to diagnose patients with PDAC.	[135]
Pancreatic cancer	Tenascin C	pancreatic fluid	upregulated	Exosome isolation is feasible from pancreatic duct fluid, and that exosomal proteins may be utilized to diagnose patients with PDAC.	[135]
Pancreatic cancer	Glypcan-1 (GCP-1)	serum	upregulated	GPC1^+^ crExos may serve as a potential non-invasive diagnostic and screening tool to detect early stages of pancreatic cancer to facilitate possible curative surgical therapy.	[137]
Pancreatic cancer	ZIP-4	cell line	upregulated	Exosomal ZIP4 promotes cancer growth and is a novel diagnostic biomarker for pancreatic cancer	[136]
Pancreatic cancer	DNA MAFs	plasma	upregulated	Exosomal DNA in combination with ctDNA provides both predictive and prognostic information relevant to therapeutic stratification	[113]
HCC/Colongiocarcinoma	Annexin V	serum	upregulated	A novel non-invasive biomarker to assess the presence and possible extent of cancers in patients with advanced liver disease	[150]

**Table 4 cancers-15-01924-t004:** Clinical relevance of tumor-educated platelets in GICs.

Cancer Type	Sample Size (Number)	Sensitivity	Specificity	AUC	Clinical Significance	References
Gastric cancer	904				NLR is better to predict overall survival than PLR in gastric cancer patients	[177]
Stage I to III liver, stomach, pancreas, and esophagus	1005	69–98%	99%		CancerSEEK localized cancer to a small number of anatomic sites in a median of 83% of the patients	[178]
Pancreatic cancer	42			82.70%	Discriminate between patients with early-stage cancer and healthy individuals	[179]
Pancreatic cancer	4				Platelet proteome can be mined for potential biomarkers of cancer.	[180]
Pan cancer (colorectal cancer, pancreatic cancer, hepatobiliary cancer)	90		81%, 71%, 58%	0.996, 0.999, 1.00	Provides a valuable platform that could potentially enable clinical advances in blood-based liquid biopsies	[169]
Liver cancer	127		96		Provides a valuable platform that could potentially enable clinical advances in blood-based liquid biopsies	[169]
Colorectal cancer	35			0.893	Differences between cancer and control samples in this study, although statistically significant, were not clinically significant	[181]

**Table 5 cancers-15-01924-t005:** Observational study with ongoing clinical trials of LB in GICs.

	Liquid Biopsy	Status	Cancer	Study Title	Study Type	Clinical Trial Identifier	Estimated Enrollment	Conditions	Interventions	Locations
CTC-1	CTC	Recruiting	Gastric cancer	Detection of CTC in the Diagnosis of Metastasis in Gastric Cancer	Observational	NCT05208372	200	Stomach Neoplasms, Metastasis	Diagnostic test: CTC test	Liaoning, China
CTC-2	CTC	Recruiting	Gastric cancer	Tumor Cell and DNA Detection in the Blood, Urine and Bone Marrow of Patients With Solid Cancers	Observational	NCT02838836	120	Esophageal Cancer, Gastric Cancer, Pancreatic Cancer, Hepatocellular Cancer, Colorectal Cancer	Procedure: study sample collection	Missouri, United States
CTC-3	CTC	Recruiting	Gastric cancer	Tumor Cell and DNA Detection in the Blood, Urine, and Bone Marrow	Observational	NCT03551951	320	Esophageal Cancer, Gastric Cancer, Pancreatic Cancer, Hepatocellular Cancer, Colorectal Cancer	Diagnostic test: test for circulating tumor cells, DNA alterations	Missouri, United States
CTC-4	CTC	Recruiting	Pancreatic cancer	Heat Shock Protein (HSP) 70 to Quantify and Characterize Circulating Tumor Cells (HSP70CTC)	Observational (Patient Registry)	NCT04628806	120	Pancreatic Cancer Stage IV	Diagnostic Test: CTC isolation by HSP70	Berlin, Germany
CTC-5	CTC	Recruiting	Liver cancer	Prognostic Value of Liver Cancer CTCs Isolated by a Novel Microfluidic Platform	Observational (Patient Registry)	NCT05242237	300	Hepatocellular Carcinoma, Circulating Tumor Cell, Whole Genome Sequencing		Chongqing, China
CTC-6	CTC	Recruiting	Liver cancer	Clinical Study for Combined Analysis of CTC and Exosomes on Predicting the Efficacy of Immunotherapy in Patients with Hepatocellular Carcinoma	Observational	NCT05575622	200	HCC	Device: CTC PD-L1, exosomal PD-L1, and exosomal LAG-3 detection	Hubei, China
CTC-7	CTC	Recruiting	Liver cancer	The Role of Circulating Tumor Cells As Markers of Advanced Disease and Prognosis In HCC	Observational	NCT04800497	200	Hepatocellular Carcinoma, Recurrent Hepatocellular Carcinoma, Circulating Tumor Cell	Procedure: hepatic resection	4 locations in Italy
CTC-8	CTC	Recruiting	Colorectal cancer	Sample Collection Study for the CellMax Life Circulating Tumor Cell and Circulating Tumor DNA Platforms for the Early Detection of Colorectal Cancer and Adenomas	Observational	NCT05127096	100	Colorectal Cancer Screening	Diagnostic test: FirstSight blood test	Alabama, California, United States
ctDNA-1	ctDNA	Recruiting	Gastric cancer	Detection of ctDNA in the Diagnosis of Metastasis in Gastric Cancer	Observational	NCT05208372	200	Stomach Neoplasms, Metastasis	Diagnostic test: ctDNA test	Liaoning, China
ctDNA-2	ctDNA	Recruiting	Gastric cancer	ctDNA Screening in Advanced HER2 Positive Gastric Cancer	Observational	NCT04520295	100	HER2-Positive Gastric Cancer	Genetic: ctDNA screening	Shanghai, China
ctDNA-3	ctDNA	Recruiting	Gastric cancer	Monitoring Minimal Residual Disease in Gastric Cancer by Liquid Biopsy Study Description	Observational	NCT05029869	100	Gastric Cancer, ctDNA	Diagnostic test: ctDNA	Ho Chi Minh City, Vietnam
ctDNA-4	ctDNA	Recruiting	Gastric cancer	Potential Clinical Utilities of Circulating Tumor DNA in Advanced HER2 Negative Gastric Cancer	Observational	NCT05513144	30	Gastric Cancer, ctDNA		Jiangsu, China
ctDNA-5	ctDNA	Recruiting	Gastric cancer	Detection of Plasma Circulating Tumor DNA in Gastric Cancer	Observational	NCT05027347	200	ctDNA, Gastric Cancer	Diagnostic test: plasma circulating tumor DNA	Ho Chi Minh City, Vietnam
ctDNA-6	ctDNA	Recruiting	Gastric cancer	Clinical Utility of Circulating Tumor DNA in Gastro-Esophageal Cancer **(CURE)**	Observational	NCT04576858	1950	Esophageal Cancer, Gastric Cancer	Diagnostic test: circulating tumor DNA	Copenhagen, Denmark
ctDNA-7	ctDNA	Recruiting	Pancreatic cancer	Observational Study of ctDNA in Resectable and Borderline Resectable Pancreatic Cancer	Observational	NCT05379907	30	Pancreatic Cancer	Other: SIGNATERA™ ctDNA testing	Virginia, United States
ctDNA-8	ctDNA	Recruiting	Pancreatic cancer	ctDNA Assay in Patients with Resectable Pancreatic Cancer	Observational	NCT05052671	50	Pancreas Cancer		Oklahoma, United States
ctDNA-9	ctDNA	Recruiting	Pancreatic cancer	Liquid Biopsy for ctDNA in Peritoneal Lavage and Blood in Pancreatic Cancer	Observational (Patient Registry)	NCT05400681	200	Pancreatic Cancer, Pancreatic Adenocarcinoma		Odense, Denmark
ctDNA-10	ctDNA	Recruiting	Pancreatic cancer	Prognostic Role of Circulating Tumor DNA in Resectable Pancreatic Cancer (**PROJECTION**)	Observational	NCT04246203	200	Pancreatic Cancer	Other: liquid Biopsy	Bavaria, Berlin, Cologne, Germany
ctDNA-11	ctDNA	Recruiting	Pancreatic cancer	DNA Mutation Detection in Circulating Tumor DNA and Tissue by mmADPS for Pancreatic Cancer	Observational (Patient Registry)	NCT05604573	150	Pancreatic Cancer	Diagnostic Test: cell-free DNA in blood, genetic mutation in tissue	Seoul, South Korea
ctDNA-12	ctDNA	Recruiting	Liver cancer	Tumor Cell and DNA Detection in the Blood, Urine and Bone Marrow of Patients with Solid Cancers	Observational	NCT02838836	120	Esophageal Cancer, Gastric Cancer, Pancreatic Cancer, Hepatocellular Cancer, Colorectal Cancer	Procedure: study sample collection	Missouri, United States
ctDNA-13	ctDNA	Recruiting	Liver cancer	Cohort Study of Patients with Hepatocellular Carcinoma and Circulating Tumor DNA Monitoring of Chemoembolization (Mona-Lisa)	Observational	NCT05390112	167	Circulating Tumor DNA Hepatocellular Carcinoma Non-resectable	Biological: DNA	Rouen, France
ctDNA-14	ctDNA	Recruiting	Colorectal cancer	Comparison of Diagnostic Sensitivity Between ctDNA Methylation and CEA in Colorectal Cancer	Observational (Patient Registry)	NCT05558436	712	Colorectal Cancer	Diagnostic test: detection of ctDNA methylation	Guangdong, China
ctDNA-15	ctDNA	Recruiting	Colorectal cancer	Role of Circulating Tumour DNA Testing in Assessing for Alterations of Primary Anti-EGFR Resistance in RAS/RAF Wild-type Metastatic Colorectal Cancer Patients	Observational	NCT05051592	40	Colorectal Cancer		Singapore, Singapore
ctDNA-16	ctDNA	Recruiting	Colorectal cancer	Circulating Tumor DNA Analysis to Optimize Treatment for Patients with Colorectal Cancer	Observational	NCT03637686	1800	Colorectal Cancer		10 locations in Denmark
ctDNA-17	ctDNA	Recruiting	Colorectal cancer	Tracking Mutations in Cell Free Tumour DNA to Predict Relapse in Early Colorectal Cancer (TRACC)	Observational	NCT04050345	1000	Colorectal Cancer		36 locations in United Kingdom
ctDNA-18	ctDNA	Recruiting	Colorectal cancer	Circulating Tumour DNA (ctDNA) as a Prognostic and Predictive Marker in Colorectal Cancer—a Pilot Study	Observational	NCT04726800	300	Colorectal Cancer		8 locations in Sweden and Norway
ctDNA-19	ctDNA	Recruiting	Colorectal cancer	Sample Collection Study for the CellMax Life Circulating Tumor Cell and Circulating Tumor DNA Platforms for the Early Detection of Colorectal Cancer and Adenomas	Observational	NCT05127096	1000	Colorectal Cancer Screening	Diagnostic test: FirstSight blood test	15 locations in United States
ctDNA-20	ctDNA	Recruiting	Colorectal cancer	Dynamic Monitoring of ctDNA Methylation to Predict Relapse in Colorectal Cancer after Radical Resection	Observational (Patient Registry)	NCT03737539	300	Colorectal Cancer, ctDNA, Surveillance, Methylation	Diagnostic test: multigene methylation detection	Shanghai, China
ctDNA-21	ctDNA	Recruiting	Colorectal cancer	Epidemiological Study to Monitor Study Participants With Resected Stage II (High Risk) or Stage III Colorectal Cancer for Circulating Tumor DNA before, during and after Their Treatment with Adjuvant Chemotherapy	Observational	NCT04813627	1500	Colorectal Cancer Stage II and III	Procedure: regular blood sample collection for ctDNA assessment	67 locations in United States
ctDNA-22	ctDNA	Recruiting	Colorectal cancer	BESPOKE Study of ctDNA Guided Immunotherapy	Observational	NCT04761783	1539	Colorectal Cancer		California, United States
exo-1	Exosomes	Recruiting	Gastric cancer	Use of Circulating Exosomal LncRNA-GC1 to Monitor Gastric Cancer	Observational	NCT05397548	700	Gastric Cancer	Diagnostic test: measurement of levels of circulating exosomal lncRNA-GC1	Beijing, China
exo-2	Exosomes	Recruiting	Pancreatic cancer	Interrogation of Exosome-mediated Intercellular Signaling in Patients with Pancreatic Cancer	Observational	NCT02393703	111	Pancreatic Cancer, Benign Pancreatic Disease		New York, United States
exo-3	Exosomes	Recruiting	Pancreatic cancer	New Biomarkers in Pancreatic Cancer Using EXPEL Concept **(PANEXPEL)**	Observational	NCT03791073	200	Oncology		Montpellier, France
exo-4	Exosomes	Recruiting	Pancreatic cancer	A Pancreatic Cancer Screening Study in Hereditary High Risk Individuals	Observational	NCT03250078	100	Pancreatic Neoplasms	Diagnostic test: MRI/MRCP	Connecticut, United States
exo-5	Exosomes	Recruiting	Pancreatic cancer	A Study of Blood Based Biomarkers for Pancreas Adenocarcinoma	Observational	NCT03334708	700	Pancreatic Cancer, Pancreatic Diseases, Pancreatitis, Pancreatic Cyst	Diagnostic test: blood draw, diagnostic test: tumor tissue collection, diagnostic test: cyst fluid	13 locations in United States
exo-6	Exosomes	Recruiting	Liver cancer	A Study of Imaging, Blood, and Tissue Samples to Guide Treatment of Colon Cancer and Related Liver Tumors	Observational	NCT03432806	80	Colon Cancer, Liver Tumor	Other: blood draws, procedure: colectomy or hepatectomy, diagnostic test: Fibroscan test	New Jersey, New York, United States
exo-7	Exosomes	Recruiting	Liver cancer	Clinical Study for Combined Analysis of CTC and Exosomes on Predicting the Efficacy of Immunotherapy in Patients with Hepatocellular Carcinoma	Observational	NCT05575622	200	HCC	Device: CTC PD-L1, exosomal PD-L1, and exosomal LAG-3 detection	Hubei, China
TEP-1	Tumor educated Platelets (TEP)	Recruiting	Gastric cancer	Project **CADENCE** (CAncer Detected Early caN be CurEd)	Observational	NCT05633342	15,000	Liver Cancer, Gastric Cancer, Colorectal Cancer, Esophageal Cancer, Pancreatic Cancer		Singapore, Singapore
TEP-2	Tumor educated Platelets (TEP)	Not recruiting yet	Pancreatic cancer	ITGA2b and SELP Expression in Cancer Pancreas and Biliary Tract Cancer	Observational (Patient Registry)	NCT05493878	128	Pancreatic Cancer	Diagnostic test: mRNA expression	Assiut, Egypt
TEP-3	Tumor educated Platelets (TEP)	Recruiting	Pancreatic cancer	Pre- and Post-operative TEG Indices in Patients with or without Adenocarcinoma Undergoing Surgical Resection	Observational	NCT05517811	400	Liver Cancer, Esophageal Cancer, Colorectal Cancer, Pancreas Cancer, Biliary Cancer	Diagnostic test: TEG indices	Colorado, United States

**Table 6 cancers-15-01924-t006:** Interventional study with ongoing clinical trials of LBs in GICs.

	Liquid Biopsy	Status	Cancer	Study Title	Study Type	Clinical Trial Identifier	Estimated Enrollment	Conditions	Interventions	Locations
CTC-1	CTC	Recruiting	Gastric cancer	Liquid Biopsy in Monitiring the Neoadjuvant Chemotherapy and Operation in Gastric Cancer	Interventional (Clinical Trial)	NCT03957564	40	Gastric Cancer, Gastro-Esophageal Junction Cancer	Detection of imaging data and level of CTCs	Qinghai, China
CTC-2	CTC	Recruiting	Gastric cancer	Phase III Randomised Trial to Evaluate Folfox with or without Docetaxel (tfox) as 1st Line Chemotherapy for Locally Advanced or Metastatic Oesophago-Gastric carcinoma	Interventional (Clinical Trial)	NCT03006432	506	Esophago-Gastric Carcinoma	Drug testing and CTC level	98 locations in France
CTC-3	CTC	Recruiting	Gastric cancer	RegoNivo vs. Standard of Care Chemotherapy in AGOC	Interventional (Clinical Trial)	NCT04879368	450	Gastro-Esophageal cancer	Drug: regorafenib, biological: nivolumab, drug: docetaxel, drug: paclitaxel, drug: irinotecan, drug: trifluridine/tipracil	75 locations in United States
CTC-4	CTC	Recruiting	Gastric cancer	Avelumab + Paclitaxel/Ramucirumab (RAP) as Second Line Treatment in Gastro-esophageal Adenocarcinoma (AIO-STO-0218)	Interventional (Clinical Trial)	NCT03966118	59	Gastroesophageal Junction Adenocarcinoma, Adenocarcinoma of the Stomach	Drug: avelumab, drug: ramucirumab, drug: paclitaxel	Berlin, Germany
CTC-5	CTC	Recruiting	Gastric cancer	Ascending Doses of Ceralasertib in Combination with Chemotherapy and/or Novel Anti Cancer Agents	Interventional (Clinical Trial)	NCT02264678	330	Gastric Cancer	Drug: administration of ceralasertib in combination with carboplatin, drug: administration of ceralasertib, drug: administration of ceralasertib in combination with olaparib, drug: administation of ceralasertib in combination with durvalumab	27 locations in United States
CTC-6	CTC	Recruiting	Pancreaticcancer	Liquid Biopsy and Pancreas Cancer: Detection of AXL(+) CTCs (CTC-AXL-PANC)	Interventional (Clinical Trial)	NCT05346536	63	Pancreatic Ductal Adenocarcinoma Metastatic Pancreatic Cancer Circulating Tumor Cell	Other: detection of circulating tumor cells expressing Axl: CTC-AXL(+)	Montpellier, France
CTC-7	CTC	Recruiting	Pancreatic cancer	EUS-guided PORtal Vein Sampling for Circulating Tumor Cells in Pancreatic Cancer Patients	Interventional (Clinical Trial)	NCT05247164	70	Pancreatic Cancer, Pancreatic Adenocarcinoma	Procedure: EUS-guided portal vein sampling	Milan, Italy
CTC-8	CTC	Recruiting	Pancreatic cancer	Echo-endoscopy Biopsy Impact on the Circulating Tumor Cell Level	Interventional (Clinical Trial)	NCT04677244	42	Cancer of Pancreas	Procedure: blood sample in portal vein	Marseille, France
CTC-9	CTC	Recruiting	Liver cancer	A Trial of Adjuvant Therapy after Hepatocarcinoma Resection Based on Folate Receptor-positive Circulating Tumor Cells	Interventional (Clinical Trial)	NCT04521491	184	HCC	Drug: FOLFOX4 (infusional fluorouracil [FU], leucovorin [LV], and oxaliplatin [OXA]).	Shanghai, China
CTC-10	CTC	Recruiting	Colorectal cancer	Influence of Opioid Analgesia on Circulating Tumor Cells in Open Colorectal Cancer Surgery	Interventional (Clinical Trial)	NCT03700411	120	Colorectal Cancer, Circulating Tumor Cell	Drug: morphine, piritramid, epidural	3 locations in Czech Republic
CTC-11	CTC	Recruiting	Colorectal cancer	Tumoral Circulating Cells and Colorectal Cancer Progression	Interventional (Clinical Trial)	NCT03256084	120	Colorectal Cancer	Procedure: blood and tumor samples	Marseille, France
ctDNA-1	ctDNA	Recruiting	Gastric cancer	Liquid Biopsy in Monitoring the Neoadjuvant Chemotherapy and Operation in Gastric Cancer	Interventional (Clinical Trial)	NCT03957564	40	Gastric Cancer, Gastro-Esophageal Junction Cancer	Drug: neoadjuvant chemotherapy with PSOX regimen. Other: detect the imaging data and levels of CTC, ctDNA, cfDNA, CEA, CA19-9, CA72-4 in plasma. Other: detect the tumor-related DNA in pathological tissues after operation. Other: follow-up of DFS and OS in patients with gastric cancer after operation.	Qinghai, China
ctDNA-2	ctDNA	Recruiting	Gastric cancer	MR-guided Pre-operative RT in Gastric Cancer	Interventional (Clinical Trial)	NCT04162665	36	Gastric Adenocarcinoma	Radiation: MR-guided radiation therapy, procedure: blood for ctDNA	Missouri, United States, Seoul, South Korea
ctDNA-3	ctDNA	Recruiting	Gastric cancer	Peritoneal Carcinomatosis Leveraging ctDNA Guided Treatment in GI Cancer Study (PERICLES Study)	Interventional (Clinical Trial)	NCT04929015	30	Gastric Cancer, ctDNA	Colorectal carcinoma by AJCC V8 stage, digestive system neoplasm, esophageal carcinoma by AJCC V8 stage, gastric carcinoma by AJCC V8 stage, liver and intrahepatic bile duct carcinoma, peritoneal carcinomatosis	New Jersey, United States
ctDNA-4	ctDNA	Recruiting	Pancreatic cancer	Pilot Comparing ctDNA IDV vs. SPV Sample in Pts Undergoing Biopsies for Hepatobiliary and Pancreatic Cancers	Interventional (Clinical Trial)	NCT05497531	15	Hepatobiliary Cancer, Pancreatic Cancer, Hepatocellular Carcinoma, Cholangiocarcinoma, Ampullary Cancer, Pancreatic Carcinoma	Diagnostic test: ctDNA blood collection	California, United States
ctDNA-5	ctDNA	Recruiting	Pancreatic cancer	PLATON—Platform for Analyzing Targetable Tumor Mutations (Pilot-study)	Interventional (Clinical Trial)	NCT04484636	200	Hepatocellular Cancer, Cholangiocarcinoma, Gallbladder Cancer, Pancreatic Cancer, Esophageal Cancer, Stomach Cancer	Diagnostic test: FoundationOne^®^CDx and FoundationOne^®^Liquid	30 locations in Germany
ctDNA-6	ctDNA	Recruiting	Liver cancer	ctDNA-Directed Post-Hepatectomy Chemotherapy for Patients With Resectable Colorectal Liver Metastases	Interventional (Clinical Trial)	NCT05062317	120	Liver Metastases	Drug: leucovorin drug: 5-FLUOROURACIL, drug: oxaliplatin, drug: irinotecan, drug: capecitabine, drug: bevacizumab	Texas, United States
ctDNA-7	ctDNA	Recruiting	Hepatocellular, Gastric	Risk factors of Immune—ChEckpoint inhibitors MEdiated Liver, gastrointestinal, endocrine and skin Toxicity (ICEMELT)	Interventionl (Clinical Trial)	NCT04631731	200	Gastric Cancer, Hepatocellular Carcinoma	Diagnostic test: blood screening, diagnostic test: tissue screening	New South Wales, Australia
ctDNA-8	ctDNA	Recruiting	Colorectal cancer	A Phase II Randomized Therapeutic Optimization Trial for Subjects with Refractory Metastatic Colorectal Cancer Using ctDNA: Rapid 1 Trial	Interventional (Clinical Trial)	NCT04786600	78	Metastatic Colorectal cancer	Device: Signatera ctDNA assay, drug: pre-specified sequence of FDA-approved drugs and drug combinations	Florida, United States
ctDNA-9	ctDNA	Recruiting	Colorectal cancer	Initial Attack on Latent Metastasis Using TAS-102 for ct DNA Identified Colorectal Cancer Patients after Curative Resection (ALTAIR)	Interventional (Clinical Trial)	NCT04457297	240	Colorectal Neoplasms, Trifluridine, and Tipiracil, Circulating Tumor DNA	Drug: trifluridine and tipiracil, drug: placebo	39 locations in Japan
ctDNA-10	ctDNA	Recruiting	Colorectal cancer	IMPROVE Intervention Trial Implementing Non-invasive Circulating Tumor DNA Analysis to Optimize the Operative and Postoperative Treatment for Patients with Colorectal Cancer	Interventional (Clinical Trial)	NCT03748680	64	Colorectal Cancer, Circulating Tumor DANN, Adjuvant Chemotherapy, Progression Free Survival	Drug: Capox (or FOLFOX) including fluoropyrimidine and oxaliplatin combination chemotherapy	4 locations in Denmark
ctDNA-11	ctDNA	Recruiting	Colorectal cancer	Circulating Tumor DNA Analysis to Optimize the Operative and Postoperative Treatment for Patients with Colorectal Cancer—Intervention Trial 2 (IMPROVE-IT2)	Interventional (Clinical Trial)	NCT04084249	254	Colorectal Cancer, Colo-rectal Cancer, ctDNA, Gastro-Intestinal Disorder, Colorectal Neoplasms, Gastrointestinal Cancer, Gastrointestinal Neoplasms, Digestive System Disease, Digestive System Neoplasm, Colonic Diseases, Colonic Neoplasms, Colonic Cancer,	Diagnostic test: ctDNA-analysis, other: intensified follow-up schedule	Randers, Denmark
ctDNA-12	ctDNA	Recruiting	Colorectal cancer	Circulating Cell-Free Tumor DNA Testing in Guiding Treatment for Patients with Advanced or Metastatic Colorectal Cancer	Interventional (Clinical Trial)	NCT03844620	100	Refractory Colorectal Carcinoma, Stage III Colorectal Cancer AJCC v8, Stage IIIA Colorectal Cancer AJCC v8, Stage IIIB Colorectal Cancer AJCC v8, Stage IIIC Colorectal Cancer AJCC v8, Stage IV Colorectal Cancer AJCC v8, Stage IVA Colorectal Cancer AJCC v8, Stage IVB Colorectal Cancer AJCC v8 Stage, IVC Colorectal Cancer AJCC v8	Other: best practice, other: laboratory procedure, other: quality-of-life assessment, other: questionnaire administration, drug: regorafenib, drug: trifluridine and tipiracil hydrochloride	Texas, United States
ctDNA-13	ctDNA	Recruiting	Colorectal cancer	A Phase II Clinical Trial Comparing the Efficacy of RO7198457 Versus Watchful Waiting in Patients With ctDNA-positive, Resected Stage II (High Risk) and Stage III Colorectal Cancer	Interventional (Clinical Trial)	NCT04486378	201	Colorectal Cancer Stage II and III	Drug: RO7198457 intravenous (i.v.), other: observational group (no intervention)	53 locations in United States
ctDNA-14	ctDNA	Recruiting	Colorectal cancer	TAS-102 in ctDNA-defined Minimal Residual Disease in Colorectal Cancer after Completion of Adjuvant Chemotherapy	Interventional (Clinical Trial)	NCT05343013	15	Colorectal Cancer	Drug: TAS-102	Texas, United States
ctDNA-15	ctDNA	Recruiting	Colon cancer	Colon Adjuvant Chemotherapy Based on Evaluation of Residual Disease (CIRCULATE-US)	Interventional (Clinical Trial)	NCT05174169	1912	Stage III Colon Cancer	Device: Signatera test, Drug: mFOLFOX6 3–6 month, drug: CAPOX 3 month, drug: mFOLFIRINOX, drug: mFOLFOX6 6 month, drug: CAPOX 6 month	Pennsylvania, United States
ctDNA-16	ctDNA	Recruiting	Colon cancer	DYNAMIC-III: Circulating Tumour DNA Analysis Informing Adjuvant Chemotherapy in Stage III Colon Cancer: A Multi-centre Phase II/III Randomised Controlled Study (Protocol No: ctDNA-08)	Interventional (Clinical Trial)	ACTRN12617001566325	356/1000	Stage III Colon Cancer	Pre-operative combined chemotherapy and radiotherapy, post-operative combined chemotherapy and radiotherapy, 3 months of fluoropyrimidine adjuvant chemotherapy, ECOG performance status 0–2	NSW, NT, QLD, SA, TAS, WA, VIC, Australia
exo-1	Exosomes	Recruiting	Gastric cancer	A Study of exoASO-STAT6 (CDK-004) in Patients With Advanced Hepatocellular Carcinoma (HCC) and Patients With Liver Metastases From Primary Gastric Cancer and Colorectal Cancer (CRC)	Interventional (Clinical Trial)	NCT05375604	30	Advanced Hepatocellular Carcinoma (HCC), Gastric Cancer Metastatic to Liver, Colorectal Cancer Metastatic to Liver	Drug: CDK-004	California, New York, Tennessee, United States
exo-2	Exosomes	Recruiting	Pancreatic cancer	iExosomes in Treating Participants with Metastatic Pancreas Cancer with KrasG12D Mutation	Interventional (Clinical Trial)	NCT03608631	28	KRAS NP_004976.2:p.G12DMetastatic Pancreatic AdenocarcinomaPancreatic Ductal AdenocarcinomaStage IV Pancreatic Cancer AJCC v8	Drug: mesenchymal stromal cells-derived exosomes with KRAS G12D siRNA	Texas, United States
exo-3	Exosomes	Recruiting	Pancreatic cancer	Ultra-High Resolution Optical Coherence Tomography in Detecting Micrometer Sized Early Stage Pancreatic Cancer in Participants with Pancreatic Cancer	Interventional (Clinical Trial)	NCT03711890	75	Pancreatic Carcinoma, Pancreatic Intraductal Papillary Mucinous Neoplasm, Pancreatobiliary-Type	Procedure: Optical Coherence TomographyProcedure: Therapeutic Conventional SurgeryDiagnostic Test: Laboratory Evaluation	Ohio, United States
exo-4	Exosomes	Recruiting	Liver cancer	A Study of exoASO-STAT6 (CDK-004) in Patients with Advanced Hepatocellular Carcinoma (HCC) and Patients with Liver Metastases from Primary Gastric Cancer and Colorectal Cancer (CRC)	Interventional (Clinical Trial)	NCT05375604	30	Advanced Hepatocellular Carcinoma (HCC), Gastric Cancer Metastatic to Liver, Colorectal Cancer Metastatic to Liver	Drug: CDK-004	3 locations in United States
TEP-1	Tumor educated Platelets (TEP)	Recruiting	Pancreatic cancer	Serial Measurements of Molecular and Architectural Responses to Therapy (SMMART) PRIME Trial	Interventional (Clinical Trial)	NCT03878524	40	Stage II Pancreatic Cancer AJCC v8, Stage III Pancreatic Cancer AJCC v8, Stage IV Pancreatic Cancer AJCC v8, Stage IV AJCC v8, Unresectable Pancreatic Adenocarcinoma	Drug testing	Oregon, United States

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
