# Peer review of "Current Applications of Liquid Biopsy in Gastrointestinal Cancer Disease—From Early Cancer Detection to Individualized Cancer Treatment"

_cancers, 2023, doi:10.3390/cancers15071924_

Round 1
Reviewer 1 Report
The manuscript has reviewed the possible applications of liquid biopsy in the clinical management of gastrointestinal cancers.
After a minor revision, the manuscript can be accepted. I recommend the below modifications:
English language and style are fine/minor spell check required, e.g. line 12: Gastrointestinal cancers are the most common cancers,...; line 73: "small minority" --> minority
Use the abbreviation of liquid biopsy (LB) at the first occurrence, line 18, and stick to it afterward (in line 20, liquid biopsy is used again). It also applies to cell-free DNA and CRC later on in the text.
References are missing from the first paragraph of the introduction. Or all this information is from ref [1]?
Use another phrase instead of "in addition" in line 42, as it is repeated.
line 72: CTCs are not only shedding from the "edges" of the tumor tissue, please rephrase.
line 110, 111: put the references at the end of the clause.
lines 161-163 references are missing
lines 169-171 please correct the sentence, "ctDNA content reasonable prediction" is not clear here
line 201 "During Pancreatic cancer" - in case of pancreatic cancer?
Table 5 Delete column 1, named "Row".
Author Response
We thank the reviewer for the positive comments and accepting the manuscript after minor modifications. We responded to your suggestion pointwise. Please see the attachment.

Reviewer 2 Report
This reviewer read this review article carefully and found this, Table 1 and TEP in particular, to be meaningful and of high quality for readers caring for GI patients.
Although the main text describes liquid biopsy including TEP in the liver, gallbladder, and pancreas cancers, they are not mentioned in the abstract and may need to be sorted out.
Overall, it would be more informative if the characteristics of LB in GI cancers are discussed in comparison to cancers of other organs.
Author Response
We thank the reviewer his his appreciation towards our manuscript. We have replied to your suggestion point by point. Please see the attachement.
